

# Feldspathic sandstone as an emerging soil stabilizer for aeolian sand in the Mu Us Sandy Land: insights into particle size distribution

Lu Zhang[1,2,3,4,5], Jichang Han[1,2,3,4,5], Juan Li[1,2,3,4,5], Shenglan Ye[1,2,3,4,5] and Dan Wu[1,2,3,4,5]

[1] Technology Innovation Center for Land Engineering and Human Settlements, Xi'an, China
[2] Shaanxi Provincial Land Engineering Construction Group Co., Ltd., Xi'an, China
[3] Key Laboratory of Degraded and Unused Land Consolidation Engineering, Ministry of Natural Resources, Xi'an, China
[4] Shaanxi Engineering Research Center of Land Consolidation, Xi'an, China
[5] Land Engineering Technology Innovation Center, Ministry of Natural Resources, Xi'an, China

Corresponding authors
Lu Zhang, luluqiaofeng@126.com
Jichang Han, 2918173256@qq.com

## ABSTRACT

Stabilization of aeolian sand is essential for achieving desertification control, soil and water conservation, and agricultural development in sandy lands. Feldspathic sandstone is a soft clay rock widely found in the Mu Us Sandy Land. The purpose of this study was to ascertain the mechanism for aeolian sand stabilization with feldspathic sandstone from the perspective of particle size distribution. Feldspathic sandstone was added to aeolian sand at different ratios ($m_f$:$m_s$ = 1:0, 1:1, 1:2, 1:5, and 0:1, where $m_f$ is the mass of feldspathic sandstone and $m_s$ is the mass of aeolian sand). The results showed that the soil texture was modified upon addition of feldspathic sandstone. The content of particles <0.05 mm increased with increasing addition ratio of feldspathic sandstone, in contrast to the downward trend observed for particles >0.05 mm. Consequently, the soil texture changed from sand to sandy loam, then loam, and finally silty loam. The addition of feldspathic sandstone ameliorated aeolian sand, resulting in a broader particle size distribution and lower particle size uniformity. Continuously well-graded soil was obtained at $m_f$:$m_s$ = 1:5 (coefficient of uniformity: 54.71; coefficient of curvature: 2.54) or 1:2 (coefficient of uniformity: 76.21; coefficient of curvature: 1.12). While the addition of feldspathic sandstone solved the problem of single particle size distribution in aeolian sand, the presence of aeolian sand prevented soil compaction caused by the high clay content of feldspathic sandstone. Findings of this study indicate that the addition of feldspathic sandstone to aeolian sand leads to the mixing of various sized particles and continuous gradation of the soil. Although a higher addition ratio of feldspathic sandstone is more favorable for soil texture improvement, $m_f$:$m_s$ = 1:5 is recommended for practical application in terms of particle gradation and cost effectiveness.

## INTRODUCTION

Desertification is a crucial and difficult problem in global ecology, arising from human activities and climate change. The human causes that lead to desertification mainly include inappropriate land use (*e.g.*, deforestation, unforeseen reclamation, overgrazing) and the loss of land productivity due to agricultural land occupation by mobile sand dunes (*Huang et al., 2020*). Currently, desertified soils are spreading globally at an annual rate of $5 \times 10^4$–$7 \times 10^4$ km$^2$, with more than 1 billion people and 40% of the Earth's land surface impacted by desertification. Desertified lands are mainly concentrated in arid and semi-arid areas (*Wang, 2024*).

The Mu Us Sandy Land is a typical synclinal sedimentary desert zone with a total area of $4.22 \times 10^4$ km$^2$, which epitomizes the many desertified lands in the world (*Han, Xie & Zhang, 2012*). It is located in the border triangle of Yikezhao League in Inner Mongolia Autonomous Region, Yulin City in Shaanxi Province, and Yanchi County in Ningxia Hui Autonomous Region, China. In this region, the land suffers from heavy desertification due to wind erosion, with infertile soil and scarce water resources in a vulnerable ecological environment and is one of the most seriously desertified areas in northern China and a major source of sandstorms in the Beijing–Tianjin region of China. As early as 1934, *Cheng (1934)* published an article ('*Desert expansion in northern China*'), which described the desertification process on the southern edge of the Mu Us Sandy Land and highlighted the fact of continuous desert expansion. This issue has also been documented in recent years (*e.g.*, *Wang & Zhao, 2005*). Importantly, the Mu Us Sandy Land is one of the most solar-rich (light resource is abundant, and ground crops can receive more light and heat resources) areas in China and a high-quality production area for potato (*Solanum tuberosum* L.) and maize (*Zea mays* L.). Therefore, the consolidation and development of desertified lands exemplified by the Mu Us Sandy Land is of double significance to the regional environment and agriculture.

Desertification can be combated through afforestation by aerial sowing (*Shen, 1998*; *Liu et al., 2020*), setting apart sand areas for tree and grass growing (*Cheng et al., 2018*), and restoration with ecological water conservancy (*Lu et al., 2018*; *Cao, 2011*; *Cao et al., 2011*). However, these traditional soil and water conservation strategies are costly and need to be implemented for at least decades. As aeolian sand is most deficient in clay, its uniform and loose texture hinders the formation of a stable soil structure. Hence, aeolian sand is prone to water infiltration, nutrient loss, and soil erosion, resulting in land infertility and barrenness. Strategies for the amelioration of aeolian sand often involve the addition of clay or organic amendment. In particular, the role of clay as a root inducer in stabilizing aeolian sand has received global attention (*Kravchenko et al., 2015*; *Müller & Höper, 2004*). While the available methods of sandy soil improvement are disadvantageous in the transportation of materials from the source region, the cost of large-scale desertification control projects can be saved by using local materials in the desertified land.

Feldspathic sandstone is a soft clay rock distributed over a large area ($\sim$$1.67 \times 10^4$ km$^2$) in the Mu Us Sandy Land. It consists of thick sandstone interbedded by sandy shale and argillaceous sandstone, with low diagenetic degree, high weathering potential, and

weak intergranular cementation. This rock is hard without water and soft in contact with water. Of course, the "hard" and "soft" here refer to whether the feldspathic sandstone can be broken or crushed by hand in the wild (*Wang et al., 2007*). It is the main source of coarse sand in the Yellow River, known as "Earth's ecological cancer". Despite being an object of land governance, feldspathic sandstone resources have not yet been exploited and utilized. Based on the knowledge of soil erosion mechanisms and the practice of sandy land governance, the occurrence of soil erosion is in close association with the particle size distribution of surface soil (*Li, Shen & Xie, 2009*; *Pope & Odhiambo, 2014*; *Sadeghi, Harchegani & Asadi, 2017*; *Zhang & Han, 2019*; *Gao et al., 2021*; *Han, Li & Yin, 2021*). As feldspathic sandstone and aeolian sand are respectively compacted and loose in structural property, the use of feldspathic sandstone as a sandy soil stabilizer is expected to reverse the traditional idea of controlling desertification to promoting soil formation. Previous studies have demonstrated improvements in the water retention capacity, nutrient conditions, and crop yields of aeolian sand stabilized with feldspathic sandstone (*Liang et al., 2019*; *Li et al., 2019*; *Sun & Han, 2018*; *Zhen et al., 2016*; *Li, Rao & Xu, 2022*; *Li, Zhang & Yu, 2022*; *Hu et al., 2023*; *Zheng, Dang & Xue, 2023*). Presently, the mechanism driving the improvement of aeolian sand with feldspathic sandstone has not been reported from a particle size distribution perspective. Particle size distribution is a single index, which makes it difficult to establish a system, resulting in less relevant research. Moreover, because soil improvement is a long process, especially for the improvement of particle size distribution, the research period is long and the scientific research cost is high, which is also one of the reasons for the lack of relevant researches at present. In order to solve this problem, we adopt the method of pilot test in the experimental area, and on the basis of laboratory experimental research, further determine the change of particle size composition after the combination of feldspathic sandstone and aeolian sand soil, which lays a foundation for field popularization and application in the later stage.

The fundamental guarantee of national food security lies in cultivated land, and land renovation is an important means to supplement cultivated land, which will effectively ensure that "Chinese people put their rice bowls in their own hands". Faced with a research gap, in this study, we analyzed the changes of particle size distribution and the patterns of particle size distribution in aeolian sand after mixing with feldspathic sandstone at various ratios. Particle size was used as an indicator to unravel the mechanisms of aeolian sand stabilization with feldspathic sandstone and explore the farmland consolidation model for eco-environmental conservation based on the trinity of quantity, quality, and ecology. The results of this study could be useful to compensate for the loose structure of aeolian sand, enhance the water holding capacity of the composite soil, improve water resource utilization efficiency, control soil erosion, and support the application of feldspathic sandstone in water conservation, soil erosion control, and agricultural development in sandy lands.

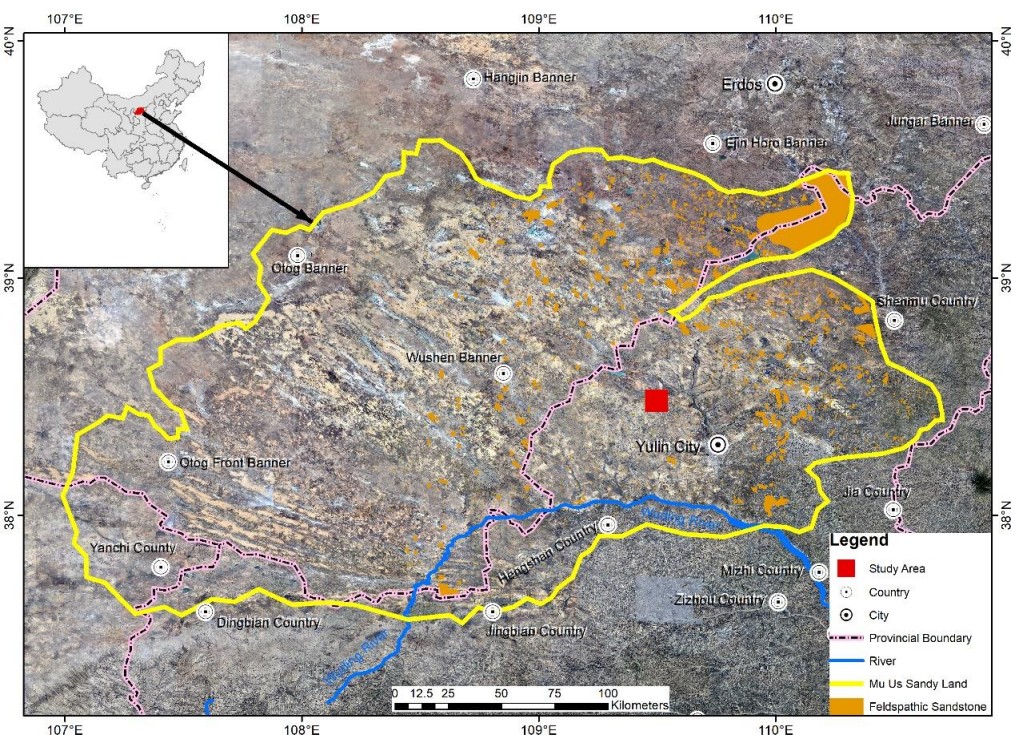

**Figure 1** **Location of the study area in the Mu Us Sandy Land.** The remote sensing image was taken in the northwest of Yulin City, Shaanxi Province, China, located at the junction of Shaanxi, Ningxia and Inner Mongolia. Image and data source: Geospatial Data Cloud Platform of Computer Network Information Center of Chinese Academy of Sciences (http://www.gscloud.cn).

## MATERIALS AND METHODS

### Study area

In 2012, we initiated the engineering construction and established an experimental field at the sandy land consolidation project site in Dajihan Village, Xiaogihan Township, Yulin District, Yulin City, northern Shaanxi Province. The study area (109°28′58″–109°30′10″E, 38°27′53″–38°28′23″N; Fig. 1) is located on the southern edge of the Mu Us Sandy Land and in the middle reaches of the Wuding River basin, with an elevation of 1206–1215 m. This area belongs to a typical mid-temperate semi-arid continental monsoon climate zone characterized by four distinct seasons. It has a long winter and a short summer, with a windy and dry spring and a cool and humid autumn (*Zhang et al., 2024*).

The mean annual temperature of the study area is 8.1 °C and its ≥10 °C cumulative temperature is 3307.5 °C. The lasting days of ≥10 °C cumulative temperature and mean annual frost-free period are 168 and 154d, respectively. The annual precipitation varies from 250 to 440 mm (mean: 413.9 mm), and 60.9% of the precipitation is concentrated in July–September with rainy and hot weather. The extreme annual precipitation has a maximum of 695.4 mm and a minimum of 159.6 mm, and the maximum daily precipitation reaches 141.7 mm. The mean annual sunshine duration is 2879 h and the percentage of sunshine is 65% per year. The total annual radiation is 606.94 kJ/cm$^2$ and the dryness is

between 1.0–2.5. Sand-driving wind with speeds >5 m/s occurs 220–580 times per year (The height at which the wind speed has been recorded was 11 meter), and sand dunes are <10 m in height (*Sun, Ma & Cao, 1995*).

## Experimental design and setup

We adopted an engineering approach (Fig. 2) for soil stabilization (the process flow includes land leveling, roll compaction, paving feldspathic sandstone, soaking feldspathic sandstone, mixing and roll compaction) and plot construction with five different ratios of feldspathic sandstone to aeolian sand ($m_f$:$m_s$ = 1:0, 1:1, 1:2, 1:5, and 0:1, where $m_f$ is the mass of feldspathic sandstone and $m_s$ is the mass of aeolian sand). Before mixing feldspathic sandstone and aeolian sand, feldspathic sandstone needs to be transported and broken, which means that feldspathic sandstone is broken into pieces ≤4 cm by hand or hammer and kept for mixing. The depths to which feldspathic sandstone was added ranged from 8 to 12 centimetre. Each treatment was applied to one experimental plot (20 m long × 7 m wide) with three replications, and a total of 15 plots were set up.

In the northern Shaanxi region of China, the climate is semi-arid climate, four distinct seasons, long light time, suitable temperature, moderate annual precipitation. This climatic condition makes northern Shaanxi suitable for growing potatoes, because potatoes need sufficient light and temperature to grow, and potatoes also need a certain amount of rainfall. These climatic conditions can be met in northern Shaanxi, so the natural environment for growing potatoes is very superior. Moreover, potatoes are popular in the local market and the market demand is large. So from early April 2013, potato (*Solanum tuberosum* L. cv. Longshu-7) was grown in the experimental field with a row spacing of 70 cm, a plant spacing of 25 cm, and one cropping season per year. Potato yield of the 10th season was estimated after harvest at the end of September 2023. At the time of harvest, surface soil samples (0–20 cm depth) were collected from five random points in each plot. The soil samples were air-dried, ground, and sieved through a 2-mm mesh before the analysis of particle size and the determination of physicochemical and mechanical properties.

To verify the soil improvement after stabilization, we randomly collected five loessial soil samples (0–20 cm depth) in potato field adjacent to the project site based on the soil type map of China from the second national soil census (2023). The physicochemical and mechanical properties of loessial soil samples were analyzed and compared with the properties of stabilized soil samples.

## Sample analysis

Grain size of the experimental soils was analyzed by laser diffraction using the Mastersizer 3000 laser particle size analyzer (Mastersizer 3000; Malvern Instruments Ltd., Worcesteshire, UK). The range of grain-size distribution was determined based on the Chinese system of grain size fractionation (*Huang, 2005*), and soil mechanical distribution was analyzed based on the USDA soil texture ternary diagram (USDA; *Huang, 2005*). The sample dosage was 0.5 g each time, and three replicates were performed for each treatment.

Water-stable aggregates were determined using the wet-sieving technique (*Liu et al., 2016*). Capillary porosity was calculated from soil water content (measured by oven

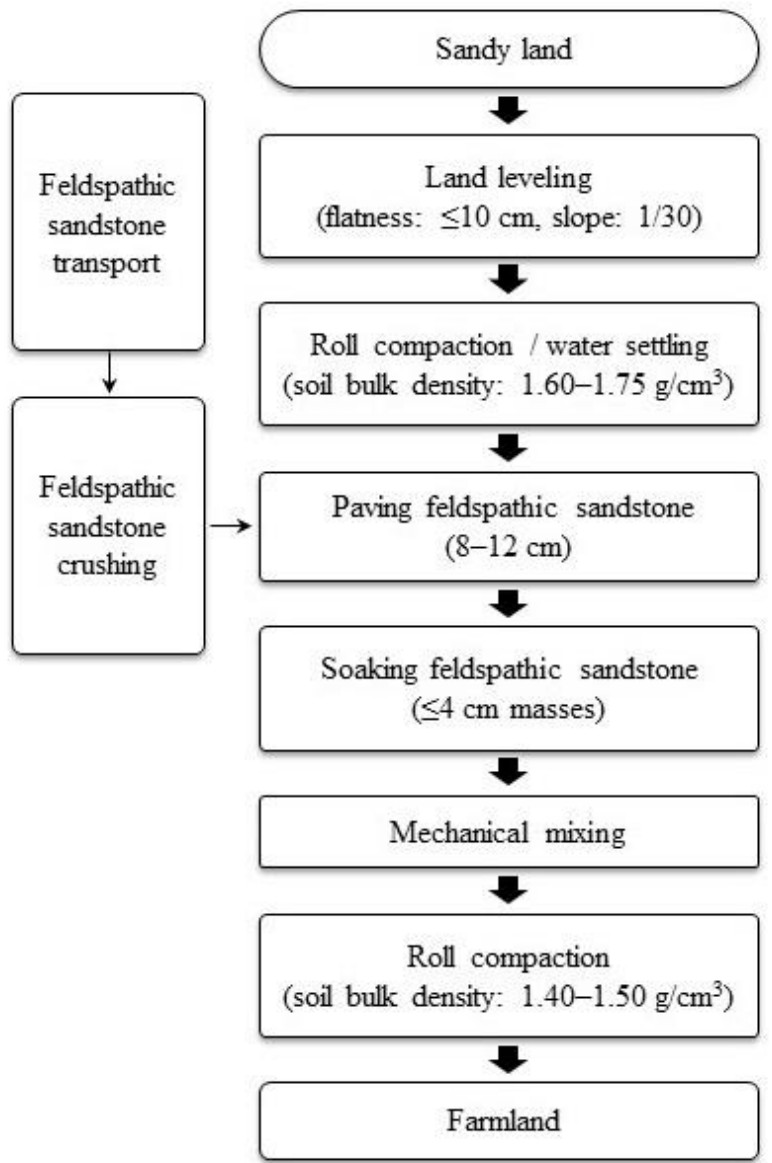

**Figure 2** **Flowchart of soil stabilization and plot construction in the experimental field.**

drying at 105 °C for 12 h) and bulk density (measured using a 100-cm³ cutting ring) (*Huang, 2005*). The determination of organic matter content and cation exchange capacity was based on oil bath heating–$K_2Cr_2O_7$ volumetric method and ammonium acetate extraction–Kjeldahl distillation method, respectively (*Huang, 2005*). Field capacity measurements were conducted using a 100-cm³ cutting ring (*Wilcox, 1962*). A four-point pattern hydraulic conductivity meter (Japan, DIK-4012) was used to measure saturated hydraulic conductivity, the oven-drying method (the SOP number is HJ 613-2011)was employed to determine effective water content, and a triaxial apparatus was used to measure the cohesion of particles and the angle of internal friction (*Huang, 2005*). The aluminum

box with the lid open (the lid is placed on the side of the aluminum box or the lid is placed flat under the box), and the oven is heated at 105 °C ± 2 °C for 8–12 h. When the oven temperature drops to about 50 °C, cover the lid, place the aluminum box in the dryer for about 30 min, cool to room temperature, and weigh. Then, open the lid of the box, and bake for 4 h, cool and weigh until the difference between the two weighing is not more than 1%.

## Data analysis

To evaluate soil gradation, the coefficient of uniformity ($C_u$) and the coefficient of curvature ($C_c$) were calculated (*Gong, Wang & Zhou, 2002*). $C_u$ measures the uniformity of soil particles ($C_u = d_{60}/d_{10}$, where $d_i$ are the particle size corresponding to the cumulative mass fraction of i%). In principle, $C_u$ is greater than 1 and a value closer to 1 indicates that the soil sample is more uniform; a soil with $C_u$ <5 is considered uniform and poorly graded. The greater the $C_u$ value, the broader the particle size distribution; a soil with $C_u$ >10 is considered well-graded. However, if the $C_u$ value is too large (generally >100, with difference in the order of magnitude), it means intermediate particle sizes may be absent, and the soil is gap graded, which means that whether the soil contains particles in various particle size ranges, if all, the soil gradation is continuous; if the soil particles in some particle size ranges are missing, the soil gradation is discontinuous and the soil is gap graded. This necessitates the use of $C_c$, which describes slope continuity in the cumulative particle gradation curve ($C_c = d_{30} \times d_{30}/(d_{60} \times d_{10})$), to evaluate soil gradation characteristics.

The experimental data are presented as means ±standard deviation ($n = 3$). SigmaPlot v13 (Systat Software Inc., San Jose, CA, USA) was used for graph construction. SPSS Statistics v18.0 (SPSS Inc., Chicago, IL, USA) was used for one-way analysis of variance followed by the Duncan's new multiple range test. A $P$-value of less than 0.05 was considered to indicate statistical significance.

## RESULTS

### Soil textural characteristics

The distribution of particles across different size ranges in soil samples with various ratios of feldspathic sandstone to aeolian sand is shown in Fig. 3. In the feldspathic sandstone ($m_f$:$m_s$ = 1:0), coarse silt (0.01–0.05 mm) accounted for the highest proportion (by mass), followed by fine silt (0.002–0.005 mm) and medium silt (0.01–0.005 mm) in comparable proportions. The feldspathic sandstone had relatively low contents of coarse clay (0.001–0.002 mm) and fine clay (<0.001 mm), with almost no particles >0.25 mm. In the aeolian sand ($m_f$:$m_s$ = 0:1), coarse sand (0.25–1 mm) was the most abundant fraction, followed by fine sand (0.05–0.25 mm). The aeolian sand had low silt content (0.05–0.002 mm, 4.05%) and almost no clay content (<0.002 mm, <1%).

In the stabilized soils containing both feldspathic sandstone and aeolian sand ($m_f$:$m_s$ = 1:1, 1:2, and 1:5), coarse sand (0.25–1 mm) comprised the largest proportion, followed by fine sand (0.05–0.25 mm) and coarse silt (0.0–0.05 mm). Taking the particle size of 0.05 mm as a boundary (which divides sand and silt fractions in the USDA soil texture), the content of particles with a size <0.05 mm in the stabilized soils exhibited a upward

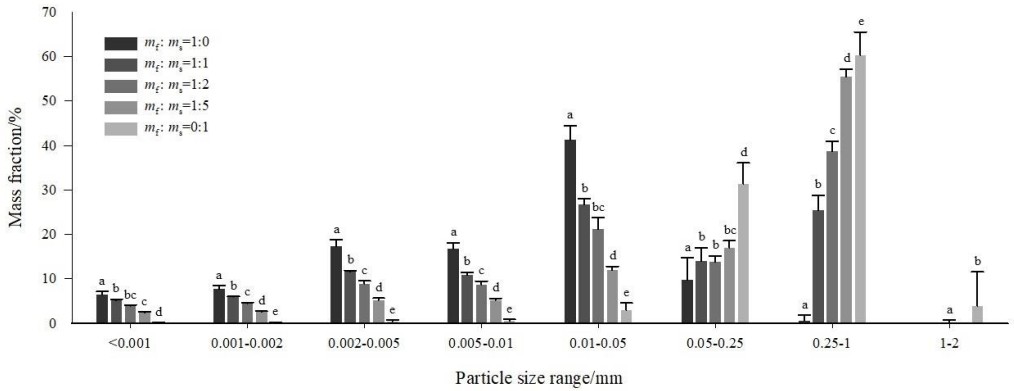

**Figure 3** **Distribution of particles across different size ranges in soil samples with various ratios of feldspathic sandstone ($m_f$) to aeolian sand ($m_s$).** Error bars represent the standard deviation of the mean, and different lowercase letters above the error bars indicate significant differences among the treatments ($P < 0.05$ by Duncan's test).

**Table 1** **Granulometric composition and texture of soil samples with various ratios of feldspathic sandstone ($m_f$) to aeolian sand ($m_s$).**

| Mass ratio ($m_f$:$m_s$) | Soil granulometric composition (%) | | | Soil texture |
|---|---|---|---|---|
| | Sand (2–0.05 mm) | Silt (0.05–0.002 mm) | Clay (<0.002 mm) | |
| 1:0 | 10.43 | 75.31 | 14.26 | Silty loam |
| 1:1 | 39.57 | 49.38 | 11.05 | Loam |
| 1:2 | 52.74 | 38.88 | 8.38 | Sandy loam |
| 1:5 | 72.55 | 22.53 | 4.92 | Sandy loam |
| 0:1 | 95.73 | 4.05 | 0.21 | Sand |

trend with increasing addition ratio of feldspathic sandstone ($m_f$:$m_s$ = 1:0 >1:1 >1:2 >1:5 >0:1), whereas the opposite pattern was observed for the content of particles with a size >0.05 mm ($m_f$:$m_s$ = 1:0 <1:1 <1:2 <1:5 <0:1). This result was attributable to the notably high proportions of >0.05 mm particles in the aeolian sand and <0.05 mm particles in the feldspathic sandstone (Fig. 3).

Table 1 provides the granulometric composition and texture of various soil samples according to the USDA textural soil classification system. With increasing addition mass ratio of feldspathic sandstone, the content of silt and clay increased in a linear pattern. The relation between feldspathic sandstone and silt contents can be expressed as: $y = 69.04x + 10.41$, $R^2 = 0.9630$; the relation between feldspathic sandstone and clay contents can be expressed as: $y = 13.37x + 2.42$, $R^2 = 0.8873$ (where $y$ is the mass fraction of soil particles, %; and $x$ is the mass fraction of feldspathic sandstone, %). The soil texture also transitioned with increasing addition ratio of feldspathic sandstone (sand–sandy loam–loam–silty loam).

## Soil gradation characteristics

The cumulative and frequency distributions of particle size distribution in various soil samples are shown in Fig. 4. The feldspathic sandstone ( $m_f{:}m_s = 1{:}0$ ) had a broad particle size distribution, with no distinct peak in the frequency distribution curve. Its cumulative particle size distribution curve also showed no steep slope, representing a polydisperse curve. These results are indicative of low particle size uniformity in the feldspathic sandstone with no dominant particle size fractions. The aeolian sand was overall coarse-grained, with particle sizes mainly ranging from 0.05 to one mm and showing a narrow peak in the frequency distribution curve. Its cumulative particle size distribution curve also showed a steep slope, representing a monodisperse curve. These results reflect that the aeolian sand had high particle size uniformity and good sorting property.

As for the stabilized soils with feldspathic sandstone and aeolian sand mixed at three different ratios ($m_f{:}m_s = 1{:}1$, 1:2, and 1:5), their frequency particle size distribution curves were divided into two portions by the particle size of 0.05 mm (Fig. 4). The content of relatively fine particles <0.05 mm increased with increasing addition ratio of feldspathic sandstone, where as the content of relatively coarse particles >0.05 mm exhibited the opposite trend. The cumulative particle size distribution curves all showed clear trend turning, which typifies polydisperse curves. However, with increasing addition ratio of feldspathic sandstone, the cumulative particle size distribution curves showed greater difference compared with the monodisperse curve of aeolian sand and shifted towards the polydisperse curve of feldspathic sandstone. This means that the uniform particle size distribution of aeolian sand was improved upon addition of feldspathic sandstone. The particle size distribution of stabilized soils showed the mixing of coarse and fine particles with broadened distribution of particle sizes.

The particle gradation parameters of soil samples with various ratios of feldspathic sandstone to aeolian sand (Table 2) were obtained based on data from laser diffraction analysis and Fig. 4. With increasing addition ratio of feldspathic sandstone, the volume average particle sizes of soils, as well as the particle sizes corresponding to various cumulative mass fractions, exhibited a downward trend. This means that the coarse-grained condition of aeolian sand was improved by adding feldspathic sandstone, and the soil particle size distribution was changed in a finer direction. The coarse grain which means 0.05–2 mm is decreasing, fine grain which means <0.05 mm is increasing. Based on the $d_i$ values of feldspathic sandstone (Table 2), the particle sizes were relatively fine and mainly concentrated in the silt and clay fractions. The feldspathic sandstone had $C_u$ value of 12.07 and $C_c$ value of 1.01. Evidence suggests that when the two conditions $C_u$ >10 and $C_c = 1$–3 are satisfied simultaneously, the samples are well graded soils (Xu et al., 2022). In summary, the feldspathic sandstone had a broad particle size distribution and continuous gradation, making it possible to serve as a soil stabilizer for aeolian sand.

Based on the $d_i$ values of aeolian sand (Table 2), the particle sizes were relatively coarse and mainly concentrated in the sand fraction. The aeolian sand had $C_u$ value of 3.32 and $C_c$ value of 1.21. Accordingly, the aeolian sand had continuous gradation with a narrow particle size distribution, and the soil was uniformly graded or poorly graded, consistent with the results presented in Fig. 4. Therefore, despite its continuous particle gradation,

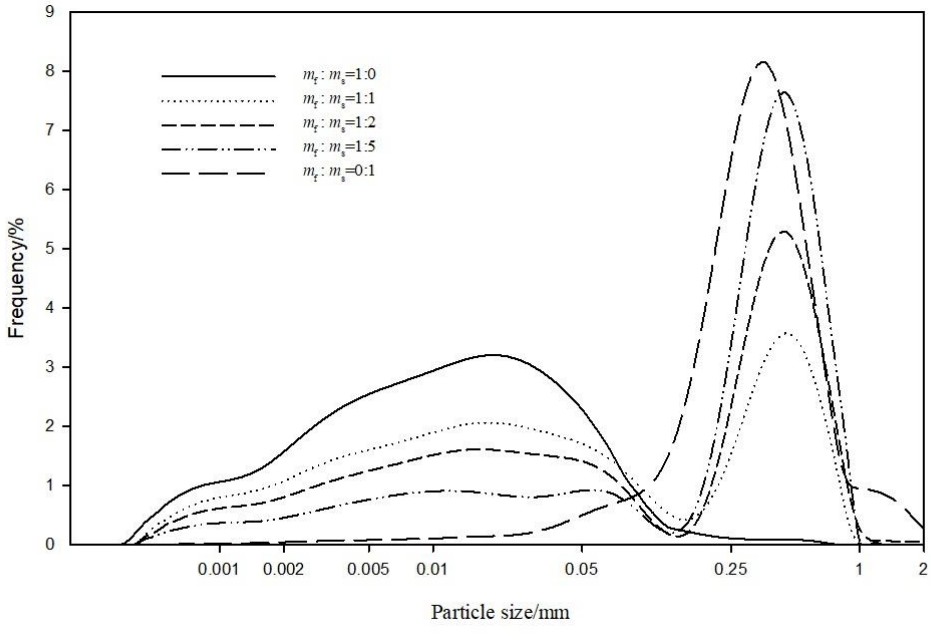

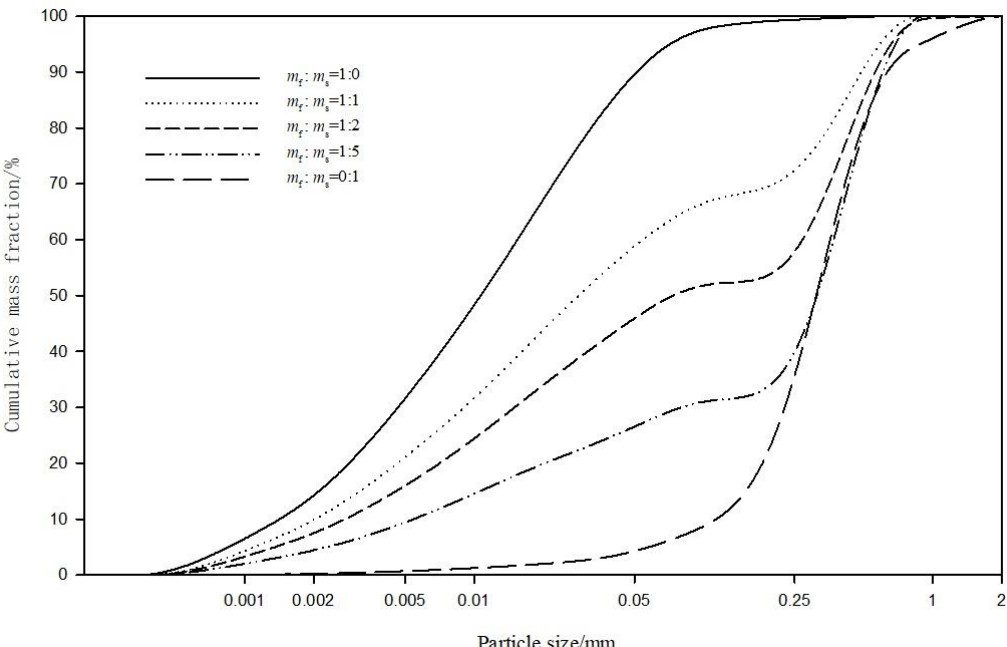

**Figure 4 The cumulative and frequency distributions of particle size composition in soil samples with various ratios of feldspathic sandstone ($m_f$) to aeolian sand ($m_s$).**

the aeolian sand had a too low $C_u$ value, a narrow particle size distribution, and lack of particles in the small size fractions, so it was classified as poorly graded soil.

Based on calculations, the $C_u$ and $C_c$ values of stabilized soils were respectively 76.21 and 1.12 at $m_f:m_s = 1:2$, and 54.71 and 2.54 at $m_f:m_s = 1:5$. In both cases, the $C_u$ values

**Table 2** Coefficient of uniformity ($C_u$) and coefficient of curvature ($C_c$) for soil samples with various ratios of feldspathic sandstone ($m_f$) to aeolian sand ($m_s$).

| Mass ratio ($m_f:m_s$) | Volume average particle size (mm) | $d_i$ (mm) | | | $C_u$ | $C_c$ |
|---|---|---|---|---|---|---|
| | | $d_{10}$ | $d_{30}$ | $d_{60}$ | | |
| 1:0 | 0.022 | 0.0014 | 0.0049 | 0.0169 | 12.0714 | 1.0148 |
| 1:1 | 0.133 | 0.0023 | 0.0096 | 0.0528 | 23.0568 | 0.7622 |
| 1:2 | 0.195 | 0.0029 | 0.0268 | 0.2210 | 76.2069 | 1.1207 |
| 1:5 | 0.267 | 0.0068 | 0.0801 | 0.3720 | 54.7059 | 2.5364 |
| 0:1 | 0.345 | 0.1125 | 0.2250 | 0.3729 | 3.3156 | 1.2071 |

**Notes.**
$d_i$ is the particle size corresponding to the cumulative mass fraction of i%.

were greater than 10 and the $C_c$ values ranged between 1–3. Therefore, the two stabilized soils showed good particle gradation at $m_f:m_s = 1:2$ or 1:5. Such soil samples had mixed composition of coarse and fine particles (*Li, Rao & Xu, 2022*; *Li, Zhang & Yu, 2022*). Furthermore, the stabilized soils had much larger $C_u$ values than feldspathic sandstone and aeolian sand (Table 2). This provides evidence that the addition of feldspathic sandstone resolved the textural defect of aeolian sand in terms of particle size uniformity, leading to superior particle size distribution and soil gradation.

## Soil physicochemical and mechanical properties

Considering the changes in soil particle size distribution after adding feldspathic sandstone to aeolian sand, we selected $m_f:m_s = 1:5$ to verify the improvements in soil physicochemical and mechanical properties and crop yield (Table 3). The addition of feldspathic sandstone changed the soil texture from sand to sandy loam after 10 seasons of potato cropping. Compared with the sandy soil, the stabilized soil showed superior properties (*e.g.*, texture, water-stable aggregates, organic matter content, cation exchange capacity) close to those of loessial soil. The potato yield of stabilized soil more than doubled that of sandy soil and reached a similar level to that of loessial soil. This indicates that following the addition of feldspathic sandstone to aeolian sand, the major soil properties and crop yield were comparable to those of loessial soil with a light loamy texture.

## DISCUSSION

The soil particle size characteristics of different land-use types in the Mu Us Sandy Land have been reported (*Wang et al., 2008*; *Chen et al., 2019*; *Mao et al., 2019*). Here, we verified the effects of adding feldspathic sandstone to aeolian sand for soil stabilization in this sandy land from a new angle—particle size distribution. As can be seen from the results in Table 1, more than 95% of aeolian sand is sand with a particle size of 0.05∼2mm, and the content of clay particles is as low as 0.2%, while the content of clay particles in feldspathic sandstone is as high as 14.3%, after the addition of feldspathic sandstone to aeolian sand, the soil was improved in texture and showed certain structural properties. The addition of feldspathic sandstone modified the coarse sandy texture of aeolian sand, offering the possibility of soil stabilization from a texture perspective. The results could provide guidance on sandy

**Table 3 Comparison of soil properties and crop yield.**

| Soil property | Sandy soil | Stabilized soil ($m_f$:$m_s$ = 1:5) | Loessial soil (reference soil) |
|---|---|---|---|
| Texture | Sand | Sandy loam | Light loam |
| ≥0.25 mm water-stable aggregates (%) | Separate particles | 20.8–29.3 | 22.9 |
| Capillary porosity (%) | 5.95 | 28.8–42.2 | 55.0 |
| Organic matter (%) | 0.09 | 0.9–1.0 | 1.0 |
| Cation exchange capacity (cmol/kg) | 3.5 | 5.0–6.5 | 6.1 |
| Field capacity ($\%_V$) | 7 | 17–38 | 24 |
| Saturated hydraulic conductivity (mm/min) | 3.41 | 0.49–1.61 | 0.93 |
| Effective water content ($\%_V$) | 2 | 13–31 | 16 |
| Cohesion of particles (Kpa) | 0 | 13–18 | 30 |
| Angle of internal friction (°) | 30–35 | 22–33 | 20 |
| Potato yield (kg/km$^2$) | 0.34 | 0.79 | 0.81 |

land consolidation, which has implications for farmland improvement and environmental protection.

In the study area, the aeolian sand showed coarse particle distribution sizes mainly concentrated in the sand fraction. The volume average particle size of aeolian sand obtained in this study is similar to that reported by Li et al. (2006). Based on the characteristics of particle size distribution and the measures of soil gradation ($C_u$ and $C_c$), the addition of feldspathic sandstone solved the problem of aeolian sand with coarse particle sizes. A higher addition ratio of feldspathic sandstone led to decrease in the sand content and increase in the silt and clay contents of the soil. This means that the particle size distribution changed from a single size range to multiple size ranges, and the soil texture became finer (sand–sandy loam–loam–silty loam). Moreover, the addition of feldspathic sandstone compensated for the limitation of aeolian sand with strong particle sorting ability, and ameliorated the poor particle gradation and uniform soil texture of sandy soil.

The changes we observed in soil particle size distribution are attributable to the abundance of silt and clay in feldspathic sandstone (Wang et al., 2007). Aeolian sand consists of >95% primary minerals (0.05–1 mm particle size), with low clay content of 0.8%, so the key problem with aeolian sand is the loss of mineral colloids. In contrast, feldspathic sandstone contains up to 16.8–46.4% secondary clay minerals (<0.002 mm particle size), with high clay content of 10.3–30.3%. Feldspathic sandstone can provide the core material—colloids—for soil formation in sandy land, remedying the defect of particle size (Zhang et al., 2019; Mohammedyasin & Wudie, 2019). Through the measurement of water conductivity, we can know that when feldspathic sandstone is added to aeolian sand, it could effectively prevent the penetration of water through the sand and negatively impact infiltration in deep soil, thereby enhancing the water retention capacity (Ma & Zhang, 2016). The soil texture varies from sand to loam with increasing content of feldspathic sandstone. This textural variation could meet the needs of crop root aeration, as well as improve soil water conditions and achieve nutrient retention (Liu et al., 2018). In terms of environmental protection, the use of feldspathic sandstone as a soil stabilizer is potentially

useful to accelerate the restoration of soils, although further testing is required on different soils and different land use contexts (*Fu et al., 2017*; *Murray, Foster & Prentice, 2012*), and as such, enhance soil resistance to wind and water erosion. With regard to agricultural production, the addition of feldspathic sandstone to aeolian sand can improve soil fertility and facilitate nutrient uptake by crop plants, leading to increase in the yield of maize (*Bouraima, He & Tian, 2016*) and potato (this study). Overall, the changes in soil particle size distribution demonstrate the potential of feldspathic sandstone as a soil stabilizer for aeolian sand.

While feldspathic sandstone plays a role in improving the texture of sandy soil, aeolian sand ameliorates poor soil aeration caused by the high clay content of feldspathic sandstone (*Li et al., 2009*). Specifically, soil clay content decreases upon addition of aeolian sand to feldspathic sandstone, and a higher addition ratio ($m_s$:$m_f$) results in lower clay content, which improved soil ventilation. Therefore, feldspathic sandstone and aeolian sand can make up for each other's limitation in soil formation and improve the particle size distribution. The addition of feldspathic sandstone to aeolian sand successfully reverses desertification into soil formation and considerably accelerates the slow process of soil formation—which usually takes 500 years to form a 1-cm thick soil (*Montgomery, 2007*; *Evans et al., 2020*; *Heimsath et al., 1997*). In this regard, it is of great significance to adopt this technique for the sustainable governance of the Mu Us Sandy Land. In addition to realizing sandy soil stabilization, this technique can increase the quantity and improve the quality of farmland.

The proposed soil stabilization technique showed the greatest effect on soil particle size distribution at $m_f$:$m_s$ = 1:2 or 1:5. Feldspathic sandstone is broken into pieces ≤4 cm which After laboratory test, it can be seen that its water retention is better, and through engineering practice, from the point of view of cost, grinding less than or equal to four cm is the most cost-effective. Taking into account the convenience of material transport and the cost of engineering projects, $m_f$:$m_s$ = 1:5 is recommended for practical application in large-scale sandy land consolidation. The economic point of view adding feldspathic sandstone to aeolian sand needs to be assessed more carefully later. The improvements in soil properties and potato yield at $m_f$:$m_s$ = 1:5 were verified by comparison with loessial soil. Figure 5 shows the distinct landscapes before and after application of the proposed technique at the land consolidation project site in an aeolian sand area along the Great Wall in northern Shaanxi Province. While this study only looked at particle size distribution, further research is needed to verify whether the recommended addition ratio of feldspathic sandstone is still optimal in terms of soil nutrients and other properties.

## CONCLUSIONS

This study verified the effect of using feldspathic sandstone to stabilize aeolian sand in the Mu Us Sandy Land. With regard to particle size, the abundance of clay in feldspathic sandstone effectively compensated for the single particle size distribution of sandy soil, making it possible to serve as a new soil stabilizer for aeolian sand. The soil texture changed from sand to sandy loam with increasing addition ratio of feldspathic sandstone. Good

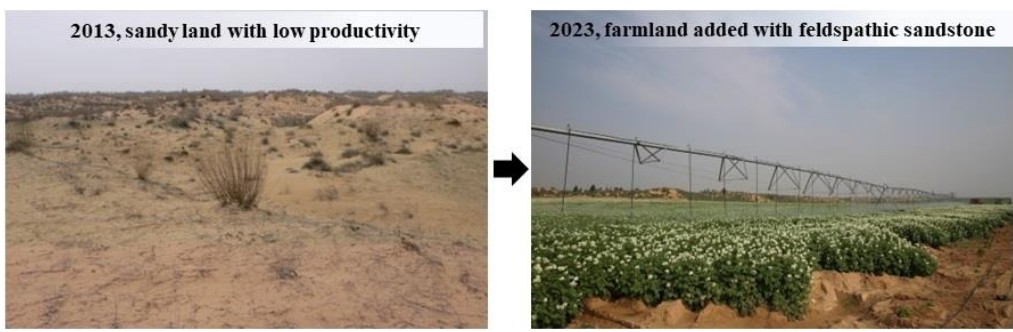

**Figure 5** **Example of landscape changes after implementing the project of "aeolian sand stabilization with feldspathic sandstone" in an aeolian sand area along the Great Wall in northern Shaanxi Province, China.** $m_f:m_s = 1:5$, photographed by the authors.

particle gradation was observed with the ratio of feldspathic sandstone to aeolian sand at 1:2 or 1:5. However, it is necessary to ascertain whether the soil is well graded with the ratio of feldspathic sandstone to aeolian sand between 1:2 and 1:5.

## ACKNOWLEDGEMENTS

The authors gratefully acknowledge researchers at the Shaanxi Provincial Land Engineering Construction Group for assistance with the field experiment.

### Funding

This research was supported by the Technology Innovation Center for Land Engineering and Human Settlements, the Shaanxi Land Engineering Construction Group Co., Ltd. and Xi'an Jiaotong University (2024WHZ0241), and the Shaanxi Provincial Innovative Talent Promotion Plan-Youth Science and Technology New Star Project (2021KJXX-88). The funders had no role in study design, data collection and analysis, decision to publish, or preparation of the manuscript.

### Grant Disclosures

The following grant information was disclosed by the authors:
Technology Innovation Center for Land Engineering and Human Settlements, Shaanxi Land Engineering Construction Group Co., Ltd. and Xi'an Jiaotong University: 2024WHZ0241.
Shaanxi Provincial Innovative Talent Promotion Plan-Youth Science and Technology New Star Project: 2021KJXX-88.

### Competing Interests

Lu Zhang, Jichang Han, Juan Li, Shenglan Ye and Dan Wu are employed by Shaanxi Provincial Land Engineering Construction Group Co. Ltd.

## Author Contributions

- Lu Zhang conceived and designed the experiments, performed the experiments, analyzed the data, prepared figures and/or tables, authored or reviewed drafts of the article, and approved the final draft.
- Jichang Han conceived and designed the experiments, authored or reviewed drafts of the article, and approved the final draft.
- Juan Li performed the experiments, prepared figures and/or tables, authored or reviewed drafts of the article, and approved the final draft.
- Shenglan Ye performed the experiments, authored or reviewed drafts of the article, and approved the final draft.
- Dan Wu performed the experiments, authored or reviewed drafts of the article, and approved the final draft.

## Data Availability

The raw data is available in the Supplemental File.

## Supplemental Information

Supplemental information for this article can be found online at http://dx.doi.org/10.7717/peerj.18577#supplemental-information.

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
