# Peer review of "Feldspathic sandstone as an emerging soil stabilizer for aeolian sand in the Mu Us Sandy Land: insights into particle size distribution"

_PeerJ, doi:10.7717/peerj.18577_

## Round 0.1 · original submission · Major Revisions

The research topic is relevant and corresponds to the direction of the journal. The submitted manuscript generally reveals the essence of the problem, and is well structured. To improve its quality, please respond to the reviewers' comments and suggestions.

Reviewer 1 ·

Basic reporting

In this study, ‘Feldspathic sandstone as an emerging soil stabilizer for aeolian sand in the Mu Us Sandy Land: insights into particle size composition’ were studied. The author analyzed the determination of the stabilization mechanism of wind sand with feldspathic sandstone from the perspective of particle size composition. Overall, this paper was developed upon an appealing topic. I found the manuscript to be well-drafted and organized. The text in many ways is clear and succinct.

Experimental design

See below

Validity of the findings

See below

Additional comments

However, I have some comments to improve the paper which you can find below.
1. Please clearly mention the novelty of your work in the introduction section.
2. Line 105: Regarding the importance of height on the wind speed rate, please add the height at which the wind speed of 5 m/s has been recorded.
3. Line 111: Please mention up to which depth the different ratios of feldspathic sandstone to aeolian sand has been applied in the experimental plots?
4. Line 144: n is 3 or 5? Please check.
5. In conclusion section, please mention the economic point of view of adding feldspathic sandstone to aeolian sand of your results or if it needs to be assessed more carefully later.
6. I suggest that some new references also be used in the paper to support the results.

Reviewer 2 ·

Basic reporting

An interesting paper on the relevant and urgent theme of desertification, but I think the paper requires significant work before it is of publishable quality. In particular, there are many instances where statements needs to be clarified, better justified, and/or evidenced with peer review literature. Many of the references are “in chinese” with no translation available/signposted meaning it is difficult for me to follow these up.

Experimental design

The methodology requires lots more detail and rationale to ensure that it is reproducible. Much of this amounts to minor additions/clarifications; collectively, however, there is a lot of information omitted currently.

Validity of the findings

I think some of the findings are over-reaching towards the end of the Discussion and Conclusion. This was a study on one soil type and one crop – so it’s a reach to say that addition of sandstone could improve for all crops. In places, statements in the Discussion do not match Results (see “additional comments” for examples).

Additional comments

L44- Isn’t climate change a cause of desertification, too?
L55- You don’t need to add the title here.
L60- What does “solar-rich” mean?
L69- What do you mean by “water leakage” – do you mean water infiltration?
L79- “Hard as stone” and “soft as mud” is quite vague. Please use more precise academic language.
L86- Compaction (or lack of it) is not a textural property. It’s a structural property.
L92- I think you need to explain the research gap.
L98- Poor texture: what about the texture makes it “poor”?
L127- You need more details about the application of sandstone to the sand. How was it applied? Was it mixed? Was it pre-treated (e.g. sieved/grounded)?
L129- Why potatoes? Please explain rationale here.
L133-You sieve through a 2 mm mesh, but you only report up to 1 mm later on. What happened to the 1-2 mm fraction in your Results? Refer to Line 185 where you report the fraction 0.25-1 mm, but not 1-2 mm.
L146- Wet-sieving. Please provide more details about this so that the experiment is reproducible.
L156- Model number of the conductivity meter?
L157- Do you have a reference/SOP for the “oven-drying method”? Once again, this is about making sure it is reproducible.
L166- Well-graded (please use the hyphen here).
L168- what do you mean by “gap graded”?
L171- Please justify the use of “means” specifically. Means are likely to be implicated by extreme values.
L204- I’d rephrase “with increasing addition ratio of feldspathic sandstone” for better clarity.
L204- Again, it seems you are using “soil texture” and “soil structure” interchangeably.
L210-212- I’d argue that this material is better placed in the Discussion. You’re starting to explain the results rather than solely describe them.
L248- “coarse-grained condition of aeolian sand was improved” – please explain what you mean here.
L260- “lacked excellent mechanical properties, its engineering properties were generally poor” – this is a little tautological.
L281- “Changed the soil texture from sand to sandy loam after 10 seasons of property cropping” – why 10 seasons? Please explain.
L296- “particle size distribution” rather than “composition”?
L298- “We conducted a detailed….” I think this can be deleted as it’s a methodological statement that you have stated previously.
L309- “Attributable to the abundance of silt and clay” – but is this the case? Refer to the results of the sandstone PSD data here. Abundance is quite a vague word.
L310- “It could effectively prevent the penetration of water through the sand” – did you empirically verify this?
L315- “The use of feldspathic sandstone as a soil stabilizer is useful to accelerate the restoration of local vegetation” – I think this may be quite a push here, given you’ve only tested this on one type of soil, with one crop.
L323- Ameliorating compaction is arguably a whole different study, and wasn’t the aim of yours here. What methods did you deploy to measure compaction, and the alleviation thereof?
L330- “slow process of soil formation” – it depends on the lithology and climate. Perhaps refer to the recent literature on soil formation/soil lifespans?
L349- “abundance of clay” – but you didn’t see much clay in the sandstone, though?
L351- “Soil texture changed from sand to loam” – earlier, you reported the change as being from “sand to sandy loam”.
L389 – Many of the references are “in Chinese”. It is difficult for me to verify these papers as suitable, as I do not have translated versions available to me.

Reviewer 3 ·

Basic reporting

The title is Feldspathic sandstone as an emerging soil stabilizer for aeolian sand in the Mu Us Sandy Land: insights into particle size composition: Experimental and Numerical Approach, the following comments have to be addressed for the paper to be in the best form to be published;


Find the attached file

Experimental design

Find the the attached file.

Validity of the findings

The authors need to add some comments to improve the work. Find the attached file.

Annotated reviews are not available for download in order to protect the identity of reviewers who chose to remain anonymous.

---

## Round 0.2 · Major Revisions

The submitted manuscript is improved. The authors took into account a significant number of comments and wishes of the reviewers. However, it needs further improvement according to the reviewers' comments.

Reviewer 1 ·

Basic reporting

the authors illustrate all of my questions

Experimental design

reasonalbale

Validity of the findings

clearly

Additional comments

no

Reviewer 2 ·

Basic reporting

I thank the authors for providing their revised version. I have read the new paper and the author responses. Please see 'Additional Comments' for line-by-line comments.

Experimental design

I thank the authors for providing their revised version. I have read the new paper and the author responses. Please see 'Additional Comments' for line-by-line comments.

Validity of the findings

I thank the authors for providing their revised version. I have read the new paper and the author responses. Please see 'Additional Comments' for line-by-line comments.

Additional comments

I have gone through each of my previous comments, and the Authors' responses. Some I am satisfied with; others require further revision before this manuscript is in a publishable state. I have pasted these below:

Reviewer #2

1. Comment: L44- Isn’t climate change a cause of desertification, too?
Response: It is mentioned in the first sentence of INTRODUCTION that "Desertification is a crucial and difficult problem in global ecology, arising from human activities and climate change." (L42-43).
Clearing that climate change is a cause of desertification. The second sentence is inaccurate, adding "human factors" and replacing it with "The human causes that…dunes." (L43-45)
Reviewer: I am satisfied with this change.

2. Comment: L55- You don’t need to add the title here.
Response: Agree. We have changed "… environment. This Mu Us Sandy Land is … region." to "… environment and is … region of China." (L54-56)
Reviewer: I am satisfied with this change.

3. Comment: L60- What does “solar-rich” mean?
Response: The “solar-rich” means that "light resource is abundant, and ground crops can receive more light and heat resources."
Reviewer: Thanks for the definition, You should make this clear in the paper.

4. Comment: L69- What do you mean by “water leakage” – do you mean water infiltration?
Response: Yes, the "water leakage" means "water infiltration". (L69)
Reviewer: I am satisfied with this change.

5. Comment: L79- “Hard as stone” and “soft as mud” is quite vague. Please use more precise academic language.
Response: We have changed this sentence to "This rock is hard without water and soft in contact with water. " (L79)
Reviewer: The terms ‘hard’ and ‘soft’ have specific meaning in geological contexts (e.g. Mohs hardness scale). Are you able to be more specific here. If the sandstone is wetted, and then dries, does its hardness return – is there a hysteresis effect here?

6. Comment: L86- Compaction (or lack of it) is not a textural property. It’s a structural property.
Response: Agree. We have changed this sentence to "As feldspathic sandstone and aeolian sand are respectively compacted and loose in structural property." (L85-86)
Reviewer: I am satisfied with this change. Perhaps a rewording to “compacted and loose, structurally.”

7. Comment: L92- I think you need to explain the research gap.
Response: Particle size composition is a single index, which makes it difficult to establish a system, resulting in less relevant research. Moreover, because soil improvement is a long process, especially for the improvement of particle size composition, the research period is long and the scientific research cost is high, which is also one of the reasons for the lack of relevant researches at present.
Reviewer: I am not sure what, if anything, has been revised in the manuscript. I’m looking for a sentence or two that explicitly describes and explains the fundamental gap in our knowledge and understanding, and how this research will address it.

8. Comment: L98- Poor texture: what about the texture makes it “poor”?
Response: We have changed "poor texture" to "loose texture". (L101)
Similar to sandy soil, whose water and fertilizer retention performance is poor.
Reviewer: As I addressed in my previous review, looseness is not a textural property. Texture refers to the size of particles. I think this should be revised to “loose structure” although, given the context of the sentence in which this is being used, perhaps you need to further explain what the result of this study is compensating, e.g. “water holding capacity, thereby addressing water conservation, soil erosion control….”


9. Comment: L127- You need more details about the application of sandstone to the sand. How was it applied? Was it mixed? Was it pre-treated (e.g. sieved/grounded)?
Response: The process flow of applying feldspathic sandstone to aeolian sand soil is added. (L129-131)
Before mixing feldspathic sandstone and aeolian sand, feldspathic sandstone needs to be transported and broken, which means that feldspathic sandstone is broken into pieces ≤4 cm by hand or hammer and kept for mixing. (L133-136)
Reviewer: Note that in the manuscript, “aeolian sand” is misspelt as “aeolite sand”. Why less than or equal to 4 cm? I’d be interested in your motivation for what is quite a coarse fraction still. You may wish to explore the scalability of this process in your Discussion, too – how can this be achieved on a large scale? – but that’s only a suggestion, rather than a recommended revision. You say “The depth different ratios were 8-12 centimetre”. I’m not sure I understand the wording here. Perhaps: “The depths to which sandstone was added ranged from 8-12 cm?”


10. Comment: L129- Why potatoes? Please explain rationale here.
Response: Because in the northern Shaanxi region of China, the climate is semi-arid climate, four distinct seasons, long light time, suitable temperature, moderate annual precipitation. This climatic condition makes northern Shaanxi suitable for growing potatoes, because potatoes need sufficient light and temperature to grow, and potatoes also need a certain amount of rainfall. These climatic conditions can be met in northern Shaanxi, so the natural environment for growing potatoes is very superior. Moreover, potatoes are popular in the local market and the market demand is large.
Reviewer: Please add this to the paper.

11. Comment: L133-You sieve through a 2 mm mesh, but you only report up to 1 mm later on. What happened to the 1-2 mm fraction in your Results? Refer to Line 185 where you report the fraction 0.25-1 mm, but not 1-2 mm.
Response: The particle size composition was determined by sieve with 2 mm aperture. Figure 3 also clearly shows the mass fraction for each particle size range (<0.001 mm、0.001–0.002 mm、0.002–0.005 mm、0.005–0.01 mm、0.01–0.05 mm、0.05–0.25 mm、0.25–1 mm and 1–2 mm). As for the results section, a selective description of figure 3 is enough, and it is not necessary to describe every particle size range.
Reviewer: I am satisfied with this change.

12. Comment: L146- Wet-sieving. Please provide more details about this so that the experiment is reproducible.
Response: Wet-sieving is a machine operation that comes with the Malvern Laser particle size analyzer itself.
To make it more accurate, we have changed it to "Grain size of the experimental soils was analyzed by laser diffraction using the Mastersizer 3000 laser particle size analyzer (Mastersizer 3000; Malvern Instruments Ltd., Worcesteshire, UK). The range of grain-size distribution was determined based on the Chinese system of grain size fractionation(Huang, 2005), and soil mechanical composition was analyzed based on the USDA soil texture ternary diagram(USDA; Huang, 2005)." (L154-158).
Reviewer: How much sample is needed for this analysis? What was your replication strategy?


13. Comment: L156- Model number of the conductivity meter?
Response: The model number of four-point pattern hydraulic conductivity meter is DIK-4012 of Japan. (L165)
Reviewer: I am satisfied with this change.


14. Comment: L157- Do you have a reference/SOP for the “oven-drying method”? Once again, this is about making sure it is reproducible.
Response: The aluminum box with the lid open (the lid is placed on the side of the aluminum box or the lid is placed flat under the box), and the oven is heated at 105 °C ±2 °C for 8-12 hours. When the oven temperature drops to about 50 °C, cover the lid, place the aluminum box in the dryer for about 30 minutes, cool to room temperature, and weigh. Then, open the lid of the box, and bake for 4 hours, cool and weigh until the difference between the two weighing is not more than 1%. (L167-171)
Reviewer: This is an interesting method – I’ve not come across this one. Does it have a SOP number? In any case, it’s good that you have inserted this into the paper, but I suggest that it isn’t parenthesized within a sentence; it should be a separate sentence.


15. Comment: L166- Well-graded (please use the hyphen here).
Response: Agree. We have changed "well graded" to "well-graded". (L180)
Reviewer: I am satisfied with this change.


16. Comment: L168- what do you mean by “gap graded”?
Response: The "gap graded" means that whether the soil contains particles in various particle size ranges, if all, the soil gradation is continuous; if the soil particles in some particle size ranges are missing, the soil gradation is discontinuous and the soil is gap graded.
Reviewer: Okay – it’s an unusual term, I haven’t seen this in the literature. Perhaps consider adding a sentence to explain this.


17. Comment: L171- Please justify the use of “means” specifically. Means are likely to be implicated by extreme values.
Response: There are some accidents and errors in one experiment. Calculating the average value of multiple experiments can reduce the experimental errors and ensure the strict accuracy of the experiment. Therefore, in the "exploration experiment", it is usually necessary to average the experimental data, the purpose is to avoid accidental factors and reduce errors.
Reviewer: Yes, but my point was why did you select “means” rather than “medians”? (Note that any data resulting from an experiment in which there were “accidents and errors” should be discarded prior to any analysis – you can’t just include these data into the average, as you’ll skew your results”.


18. Comment: L204- I’d rephrase “with increasing addition ratio of feldspathic sandstone” for better clarity.
Response: We have changed "… addition ratio of …" to "… addition mass ratio of …" (L217-218)
Reviewer: I am satisfied with this change.


19. Comment: L204- Again, it seems you are using “soil texture” and “soil structure” interchangeably.
Response: We have changed"… soil structure …" to "… soil texture…". (L218-219)
Reviewer: Okay – the sentence now states that “content of key particle size fractions (silt and clay) for soil texture formation increased in a linear pattern”. This is still difficult to understand.


20. Comment: L210-212- I’d argue that this material is better placed in the Discussion. You’re starting to explain the results rather than solely describe them.
Response: Agree. This has been moved to the first paragraph of the discussion. (L306-308)
Reviewer: Thank you, although just placing it in the Discussion isn’t sufficient. The word “this” is now out of context.


21. Comment: L248- “coarse-grained condition of aeolian sand was improved” – please explain what you mean here.
Response: The "coarse-grained condition of aeolian sand was improved" means that coarse grain (0.05–2 mm)is decreasing,fine grain (<0.05 mm)is increasing. (L262-263)
Reviewer: Check typography here – space needed within “2mm)is”. You also say “this means” and “which means” in the same sentence.


22. Comment: L260- “lacked excellent mechanical properties, its engineering properties were generally poor” – this is a little tautological.
Response: Agree. This sentence has been deleted.
Reviewer: I am satisfied with this change.


23. Comment: L281- “Changed the soil texture from sand to sandy loam after 10 seasons of property cropping” – why 10 seasons? Please explain.
Response: In the "Experimental design and setup" section, it was mentioned that after ten years (from early April 2013 to September 2023) of potato cultivation, the soil testing work of this study began. Because soil improvement is a long process, especially for the improvement of particle size composition. Therefore, the soil after ten years of planting was selected as the research improvement object in this study, which again explains question 9.
Reviewer: I am satisfied with this change.


24. Comment: L296- “particle size distribution” rather than “composition”?
Response: Agree. We have changed "… coarse particle sizes …" to "… coarse particle distribution sizes …". (L311)
Reviewer: Check whole manuscript, as there are still references to “composition”.


25. Comment: L298- “We conducted a detailed….” I think this can be deleted as it’s a methodological statement that you have stated previously.
Response: Agree. This sentence has been deleted.
Reviewer: I am satisfied with this change.


26. Comment: L309- “Attributable to the abundance of silt and clay” – but is this the case? Refer to the results of the sandstone PSD data here. Abundance is quite a vague word.
Response: More than 95% of aeolian sand belongs to primary minerals with a particle size of 0.05–1 mm, the clay content is as low as 0.8%, and the core is mineral colloid loss, while the secondary clay minerals are as high as 16.8%–46.4%, and the clay content is as high as 10.3%–30.3% of Feldspathic sandstone. Feldspathic sandstone can provide the core material - colloid for the formation of aeolian sand and make up for the defect of grain size. (L323-328)
Reviewer: Some minor English language issues to address here.


27. Comment: L310- “It could effectively prevent the penetration of water through the sand” – did you empirically verify this?
Response: Yes, we have literature and experiments to back it up.
Reviewer: What method in this study did you use to measure this? Please add to the Methods section.

28. Comment: L315- “The use of feldspathic sandstone as a soil stabilizer is useful to accelerate the restoration of local vegetation” – I think this may be quite a push here, given you’ve only tested this on one type of soil, with one crop.
Response: Yes. We carried out this study based on the project of "converting sand into soil". After ten years of planting and research, the local ecological environment has indeed been improved, which can be seen from figure 5. However, this paper only discusses the composition of soil particles.
Reviewer: Yes, but the statement is hyperbolic. I suggest a rephrasing: “The use of feldspathic sandstone as a soil stabilizer is potentially useful to accelerate the restoration of soils, although further testing is required on different soils and/or different land use contexts.”


29. Comment: L323- Ameliorating compaction is arguably a whole different study, and wasn’t the aim of yours here. What methods did you deploy to measure compaction, and the alleviation thereof?
Response: The wording is wrong. We have changed "While feldspathic sandstone … soil compaction … which reduces the possibility of soil compaction." to "While feldspathic sandstone … soil poor ventilation … which improved soil ventilation." (L341-345)
Reviewer: Soil ventilation isn’t a technical term. What do you mean here?

30. Comment: L330- “slow process of soil formation” – it depends on the lithology and climate. Perhaps refer to the recent literature on soil formation/soil lifespans?
Response: Quoting the national unified soil science teaching material of China. (L349)
Reviewer: This is a peer-reviewed journal, and therefore I strongly suggest that you use peer-reviewed literature. Rates of soil formation have been measured, and have been published in the academic literature. For example, see Montgomery (2007) (https://www.pnas.org/doi/abs/10.1073/pnas.0611508104), Evans et al. (2020) https://iopscience.iop.org/article/10.1088/1748-9326/aba2fd/meta and Heimsath et al. (1997) https://www.nature.com/articles/41056


31. Comment: L349- “abundance of clay” – but you didn’t see much clay in the sandstone, though?
Response: It can be seen from figure 3 and figure 4 that Clay content (<0.002mm) is the highest in feldspathic sandstone.
Reviewer: Satisified


32. Comment: L351- “Soil texture changed from sand to loam” – earlier, you reported the change as being from “sand to sandy loam”.
Response: This has been corrected to "sandy loam". (L372)
Reviewer: Satisified

33. Comment: L389 – Many of the references are “in Chinese”. It is difficult for me to verify these papers as suitable, as I do not have translated versions available to me.
Response: A most of the scholars in China have studied the feldspathic sandstone in the world. I can guarantee the authoritativeness and reasonableness of the quotations, whether in English or Chinese.
Reviewer: You have submitted this to an international peer-reviewed journal. It would not be scientifically responsible of me as a reviewer to take your word for it. This aside, it is also important that authors can access and read these papers outside of China.

---

## Round 0.3 · accepted · Accept

The resubmitted manuscript has been thoroughly revised in accordance with the previous comments of the reviewers, which has significantly improved its quality. The general decision of the reviewers is to accept the manuscript for publication in PeerJ. The materials are relevant and correspond to the level of the journal.

Reviewer 1 ·

Basic reporting

the authors illustrated all of my question

Experimental design

good

Validity of the findings

dood

Additional comments

no

Reviewer 2 ·

Basic reporting

No comment

Experimental design

No comment

Validity of the findings

No comment

Reviewer 3 ·

Basic reporting

Accept the revised manuscript.

Experimental design

Very Good.

Validity of the findings

No comment.